# Piezoelectricity in chalcogenide perovskites

Sk Shamim Hasan Abir [1,6], Shyam Sharma [1,6], Prince Sharma [2], Surya Karla[3], Ganesh Balasubramanian[4], Johnson Samuel [1] ✉ & Nikhil Koratkar [1,5] ✉

Piezoelectric materials show potential to harvest the ubiquitous, abundant, and renewable energy associated with mechanical vibrations. However, the best performing piezoelectric materials typically contain lead which is a carcinogen. Such lead-containing materials are hazardous and are being increasingly curtailed by environmental regulations. In this study, we report that the lead-free chalcogenide perovskite family of materials exhibits piezoelectricity. First-principles calculations indicate that even though these materials are centrosymmetric, they are readily polarizable when deformed. The reason for this is shown to be a loosely packed unit cell, containing a significant volume of vacant space. This allows for an extended displacement of the ions, enabling symmetry reduction, and resulting in an enhanced displacement-mediated dipole moment. Piezoresponse force microscopy performed on $BaZrS_3$ confirmed that the material is piezoelectric. Composites of $BaZrS_3$ particles dispersed in polycaprolactone were developed to harvest energy from human body motion for the purposes of powering electrochemical and electronic devices.

Mechanical-to-electrical energy conversion can be realized via the piezoelectric effect. Piezoelectricity is the generation of electrical charge in a solid material by an applied mechanical stress and vice versa[1]. The piezoelectric effect is exhibited in materials that develop a net electric dipole moment. In such materials, the dipole density or polarization changes in response to mechanical strain resulting in a net accumulation of charge[2]. Supplementary Table S1 lists common materials that exhibit the piezoelectric effect. The piezoelectric coefficients, $d_{ij}$ in the Table are material constants corresponding to the relation[3]: $\{S\} = [d_{ij}]\{E\}$, where S is the linearized strain matrix and E is the electric field strength. It is evident from the Table, that the highest performing piezoelectric materials such as PZT-5H, PZT-5K, PZN − 7% PT and PZNT8 all contain lead (Pb), which is a well-known carcinogen[4]. This renders these materials to be hazardous and entirely rules out biomedical applications. In fact, lead-containing materials and devices are being increasingly restricted by environmental regulations such as the Waste of Electrical and Electronic Equipment (WEEE) and Restriction of Hazardous Substances (RoHS) directives[4]. It is therefore important to

discover lead-free piezoelectric materials that offer high-performance and are comprised solely of environmentally benign, earth-abundant and non-toxic materials.

In this article, we report that chalcogenide perovskite materials exhibit piezoelectricity. Chalcogenide perovskites have the general chemical formula $ABX_3$, where A = Ba, Ca, or Sr, B = Ti, Zr, or Hf and X = S or Se (Supplementary Table S2 lists all 18 members of the family). These materials are predicted to have a direct band gap and are being widely explored in optoelectronics and photovoltaic devices[5–9]. Chalcogenide perovskites are interesting for two main reasons: (1) They are free of toxic and carcinogenic materials such as Pb and $PbI_2$; and (2) Instead of a halide ion (as in regular perovskites), a chalcogen is used (e.g., S or Se), which results in improved environmental stability[10]. While there are some recent findings[11–13] of ferroelectricity in halide perovskites and of metal-free perovskites in piezoelectric applications[14–16], there is no report to date of piezoelectricity or ferroelectricity in chalcogenide perovskites. Here, we report a pronounced piezoelectric response in $BaZrS_3$, a member of the

[1]Department of Mechanical, Aerospace and Nuclear Engineering, Rensselaer Polytechnic Institute, Troy, NY 12180, USA. [2]Department of Mechanical Engineering and Mechanics, Lehigh University, Bethlehem, PA 18015, USA. [3]Howard P. Isermann Department of Chemical and Biological Engineering and Center for Biotechnology and Interdisciplinary Studies, Rensselaer Polytechnic Institute, Troy, NY 12180, USA. [4]Department of Mechanical and Industrial Engineering, University of New Haven, West Haven, CT 06516, USA. [5]Department of Materials Science and Engineering, Rensselaer Polytechnic Institute, Troy, NY 12180, USA. [6]These authors contributed equally: Sk Shamim Hasan Abir, Shyam Sharma ✉e-mail: samuej2@rpi.edu; koratn@rpi.edu

chalcogenide perovskite family. This finding was verified both using piezoresponse force microscopy and testing of BaZrS$_3$-polymer composite films. This result is surprising, given that the stable orthorhombic phase of BaZrS$_3$ has a centrosymmetric structure. Centrosymmetric compounds inherently lack a net dipole moment and are therefore non-polar, and hence weakly piezoelectric. First-principles density functional perturbation theory (DFPT) calculations revealed that the amplified piezoelectric response observed for BaZrS$_3$, and in general for the entire family of chalcogenide perovskite materials, is attributable to its unit cell, which is comprised of many atoms, and is characterized by a significant volume of vacant space. This loosely packed structure of chalcogenide perovskites allows for an extended displacement of the ions, enabling symmetry reduction[17], and resulting in an enhanced displacement-mediated dipole moment. Such phenomenon is not as prevalent in centrosymmetric oxide perovskites such as BaZrO$_3$, which contain fewer tightly packed ions in a relatively smaller unit cell.

Having developed an understanding as to why chalcogenide perovskites exhibit piezoelectricity, we proceeded to develop practical energy harvesting devices using these materials. For this purpose, we created flexible composite films of BaZrS$_3$ particles dispersed in a polycaprolactone matrix. We successfully deployed these composite films to harvest energy from ubiquitous body movement (i.e., regular walking, jogging, running, elbow bending, clapping and hand tapping) for applications such as the charging of capacitors and the lighting of commercial light emitting diodes.

## Results

Figure 1a shows a schematic for BaZrS$_3$ powder synthesis from the BaZrO$_3$ precursor using a tube furnace and CS$_2$ as the sulfurization source (see Methods). The white BaZrO$_3$ powder turned black after sulfurization. This powder was then crushed in a mortar pestle and further characterized by X-ray Diffraction (XRD) at room temperature and ambient atmosphere. The corresponding diffractograms are reported in Fig. 1b. Sharp peaks indicates high crystallinity, and the pattern matches well with the standard reference pattern of BaZrS$_3$ (ICDD 15-0327), confirming that the sample is polycrystalline and possesses an orthorhombic distorted perovskite structure with a Pnma space group. No secondary phase is present in the sample, as no additional peaks were found, confirming complete sulfurization of the oxide precursor powder. The Rietveld refinement presented showed the cell parameters to be 7.0639 Å, 9.9809 Å, and 7.0191 Å for a, b, and c respectively. The morphology of the synthesized crystals was investigated using scanning electron microscopy (SEM) (Fig. 1b−inset), indicating the micron sized crystals to be made up of smaller nanocrystals.

Piezoelectric response of BaZrS$_3$ was characterized using conventional out of plane piezoresponse force microscopy (PFM)[18]. In PFM, a conductive atomic force microscope (AFM) cantilever probe is scanned over the sample in contact mode. While scanning the surface, an AC bias is applied to the probe tip. The electric field induces a strain in the surface, which in turn causes a periodic deflection of the cantilever, from which the effective piezoelectric charge coefficient (d$_{33,eff}$)

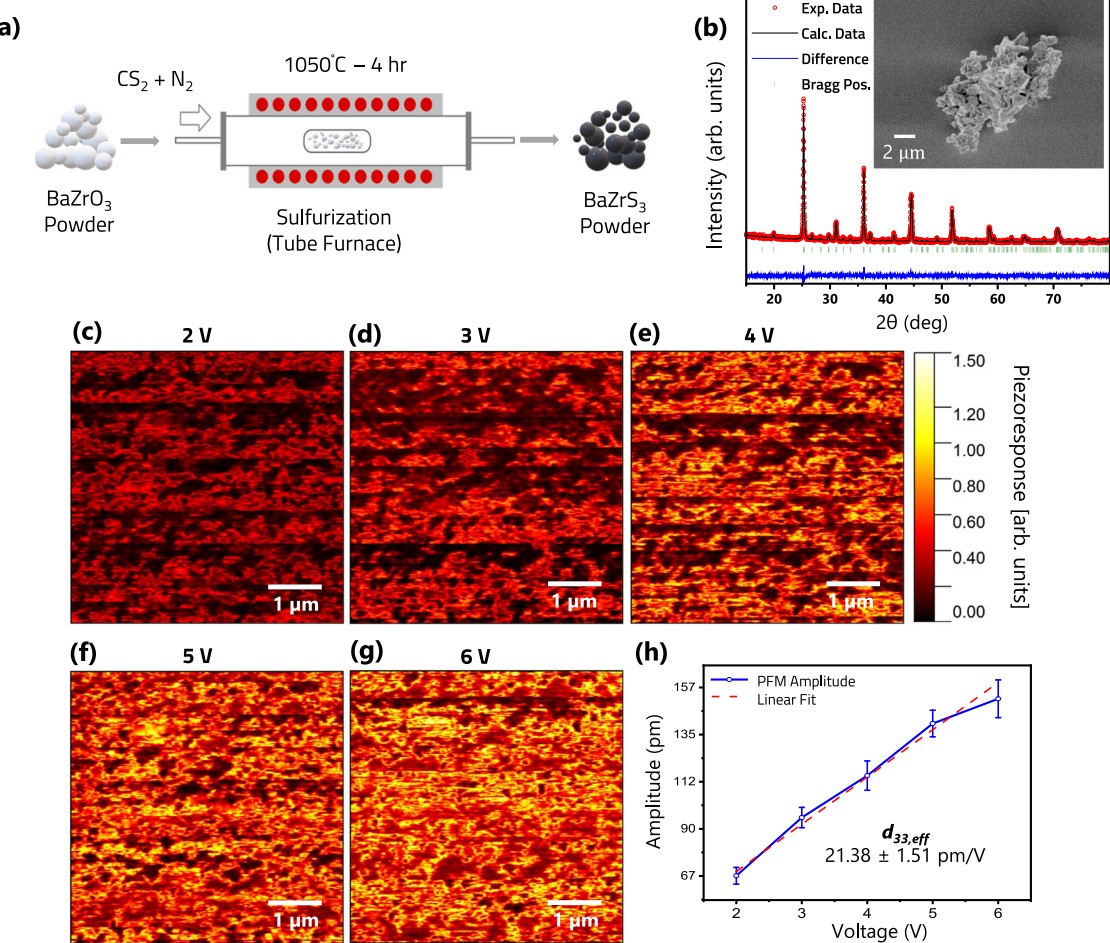

**Fig. 1 | BaZrS$_3$ perovskite synthesis and characterization. a** Schematic of BaZrS$_3$ powder synthesis. **b** XRD and Rietveld refined profile of BaZrS$_3$ powder, inset showing typical SEM image of the particle. **c**–**g** Piezoresponse force microscopy (PFM) amplitude images of BaZrS$_3$ film at different voltages from 2 to 6 V, respectively. **h** PFM amplitude plotted as a function of applied voltage. The effective piezoelectric charge coefficient (**d$_{33,eff}$**) for BaZrS$_3$ is established from the slope of the plot.

can be deduced[18]. For a randomly oriented polycrystalline material (such as BaZrS$_3$), d$_{33,eff}$ represents the overall piezoelectric response in the direction normal to the sample surface, which is also the direction of the applied electric field. For the PFM study, thin film samples of BaZrS$_3$ (~300 nm thick, Supplementary Fig. S1a) were deposited over a quartz substrate (Methods) and XRD was used to confirm the orthorhombic perovskite structure of the as-deposited film (Supplementary Fig. S1b). The BaZrS$_3$ films were cleaned using argon plasma prior to PFM characterization. To obtain PFM images, a ~5 ×5 μm$^2$ area was scanned at ~0.6 Hz scan rate, ~90° scan angle and a bias voltage was applied in the range of 0-6 V. To address the low signal to noise ratio, the probe was driven at resonance frequency (~340 kHz) and a stiff cantilever (~2.8 N/m) was used to reduce electrostatic signal contributions. For proper calibration of PFM, a periodically poled lithium niobate (PPLN) standard PFM sample from Bruker AFM Probes was utilized. Supplementary Fig. S2 shows PFM height, amplitude, and phase images of the reference PPLN sample at 2–6 V, which clearly shows the opposite polarization direction in the phase image and enhanced piezo-response amplitude with increasing voltage. PFM amplitude as a function of applied voltage was used to characterize the effective piezoelectric coefficient of the PPLN sample (Supplementary Fig. S3), which results in an d$_{33,eff}$ value of ~25.81 ± 1.87 pm/V, which is within the manufacturer specified range (25–27 pm/V).

PFM amplitude of the BaZrS$_3$ sample with increasing voltage is reported in Fig. 1c–g, which shows increasing piezo-response with increasing voltage. Figure 1h shows the piezo-response as a function of applied voltage, which results in an d$_{33,eff}$ of ~21.38 ± 1.51 pm/V. Supplementary Fig. S4 shows the PFM height, amplitude, and phase images of BaZrS$_3$ at different applied voltages. Overall, these results indicate that the chalcogenide perovskite BaZrS$_3$, exhibits a pronounced piezoelectric response with an effective piezoelectric charge coefficient that is comparable to lithium niobate, which is a well-known piezoelectric material.

## Discussion

Most high-performing piezoelectric materials are non-centrosymmetric and thus exhibit intrinsically high polarizability. In fact, this is the reason why many oxide perovskites such as BaZrO$_3$ that exhibit a centrosymmetric (cubic) crystal structure are weakly piezoelectric in their pristine (i.e., undoped) form. Centrosymmetric compounds inherently lack a net dipole moment and are therefore non-polar. In this regard, BaZrS$_3$ should in principle be no different from BaZrO$_3$, since it exhibits an orthorhombic phase which is also centrosymmetric. Why then is BaZrS$_3$ strongly piezoelectric, while BaZrO$_3$ is not? To shed light on this question, we have carried out first-principles density functional perturbation theory (DFPT) calculations. More specifically, we employed a displacement-induced polarization approach[19–21] to investigate the piezoelectric properties of the material. In this method, we induce controlled random ionic displacements of <2% (from their respective coordinates) to generate unique structural configurations and polarize the compound, without changing the volume and shape of the unit cell (see Fig. 2a–c and Methods). Subsequently, we calculate the piezoelectric stress tensor for each configuration, and average across all tensor elements. It should be noted that piezoelectric coefficients are third rank tensors, while elastic constants are fourth rank tensors. The piezoelectric stress tensors $e_{ijk} = \frac{\partial \sigma_{jk}}{\partial E_i}$ and elastic tensors $C_{lmjk} = \frac{\partial^2 U}{\partial \varepsilon_{lm} \partial \varepsilon_{jk}}$ are calculated using DFPT (see Methods) where E, σ and U are respectively the electric field, stress and strain energy[22,23]. The commonly estimated piezoelectric strain tensors, i.e., $d_{ilm} = \frac{\partial \varepsilon_{lm}}{\partial E_i}$ are calculated from $e_{ijk} = d_{ilm}C_{lmjk}$ and further simplified as $e_{ip} = d_{iq}C_{qp}$, where ε is the strain, $jk \equiv p$, $lm \equiv q$ and $i \in \{1,2,3\}$, $p \in \{1,2,3,4,5,6\}$, and $q \in \{1,2,3,4,5,6\}$[24,25].

Figure 2d, e shows the top and side views of a cubic BaZrO$_3$ (**Pm3̄m**) unit cell consisting of 5 atoms, illustrating Zirconium (Zr) at the body center (in red), Oxygen (O) at the face center (in blue) and

Barium (Ba) at the corners in green. The 3D isometric view of BaZrO$_3$ provides a pictorial replication of a densely-packed unit cell (Fig. 2f), indicating minimal space for ion displacement. The corresponding projections of the orthorhombic BaZrS$_3$ (**Pnma**) unit cell is shown in Fig. 2g–i. The unit cell of BaZrS$_3$ is comprised of twenty atoms and contains a significant volume of vacant space. This loosely packed structure of BaZrS$_3$ could in principle allow for an extended displacement of ions, thereby breaking the symmetry, and resulting in an enhanced dipole moment. Our DFPT calculations indicate that this is indeed the case (Fig. 2j). The d$_{33}$ obtained from the DFPT calculations for BaZrS$_3$ is 17.65 pC/N (or pm/V), which is comparable to many of the high-performing and non-lead containing piezoelectric materials reported in the literature (Supplementary Table S1). By contrast, BaZrO$_3$ exhibits a far lower d$_{33}$ value of −0.015 pC/N, and a maximum value of d$_{ij}$ (corresponding to d$_{22}$) of only 2.46 pC/N. The experimentally measured **d$_{33,eff}$** for BaZrS$_3$ was ~21.38 pm/V (Fig. 1h), which is comparable to the DFPT prediction. We have verified the absence of imaginary phonon modes in the phonon band structure of our specific material (Supplementary Fig. S5). The calculated band structures exhibit minimal sensitivity to the specific distorted structures employed, as evidenced by the negligible variation in phonon dispersion. This indicates that the structure is dynamically stable and highlights absence of imaginary phonons.

The results in Fig. 2 convey that the loosely packed structure and higher number of ions in the BaZrS$_3$ unit cell contributes to an amplified piezoelectric response relative to BaZrO$_3$, which consists of fewer tightly packed ions in a smaller unit cell. Thus, it is the spatial arrangement and packing of ions in the BaZrS$_3$ system that is responsible for a pronounced piezoelectric response, despite the centrosymmetric nature of the orthorhombic BaZrS$_3$ unit cell. These conclusions are extendable to the entire chalcogenide perovskite family. Supplementary Figs. S6 and S7, show DFPT calculations for CaZrS$_3$ and BaHfS$_3$. In the case of CaZrS$_3$, the maximum d$_{ij}$ is 61.74 pC/N (for d$_{14}$), which increases to 102.58 pC/N (for d$_{21}$) for BaHfS$_3$.

To develop practical energy harvesting devices using the BaZrS$_3$ material, we dispersed BaZrS$_3$ particles in a polymer (Polycaprolactone: PCL) matrix. The advantage of the polymer is that it provides ductility, which makes it relatively straightforward to mechanically deform the energy harvesting material. Note that PCL is weakly piezoelectric (~4 pC/N: Supplementary Table S1) and hence the main contributor to piezoelectricity in the composite film is the BaZrS$_3$ additives. Figure 3a depicts the solvent casting process (see Methods for details) that we used to disperse BaZrS$_3$ particles in the PCL polymer matrix. XRD testing (Supplementary Fig. S8a, b) of these PCL-BaZrS$_3$ composite films indicates that the individual PCL and BaZrS$_3$ peaks are retained in the composite, conveying that no new phase is formed, while the intensity of the BaZrS$_3$ peaks increases with BaZrS$_3$ additive concentration, as is to be expected. Figure 3b–d shows the EDS elemental map of sulfur for composite films with 5, 15 and 30 weight (wt)% BaZrS$_3$ loading. The BaZrS$_3$ particle size was quantified from these maps by calculating the maximum Feret's diameter of the boundary formed by particles which are touching and discarding any boundary <4 pixel$^2$ to take out the noise. Figure 3e–g shows the calculated boundary images and Supplementary Fig. S8c indicates that the calculated average BaZrS$_3$ particle size increases from ~0.5 to ~2 μm, with increase in BaZrS$_3$ loading from 5 to 30 weight percent. SEM images using backscattered electron detector were also taken (Fig. 3h–j) showing clear contrast between the BaZrS$_3$ particles and the PCL matrix, and particle agglomeration with increasing loading. The SEM images also show clear evidence of significant particle debonding and void formation, particularly at high BaZrS$_3$ loading.

The piezoelectric performance of the fabricated polymeric films was investigated under a range of applied pressures as shown schematically in Fig. 4a. To contact the film, copper (Cu) tape electrodes were attached to the top and bottom surfaces of the polymer films. A

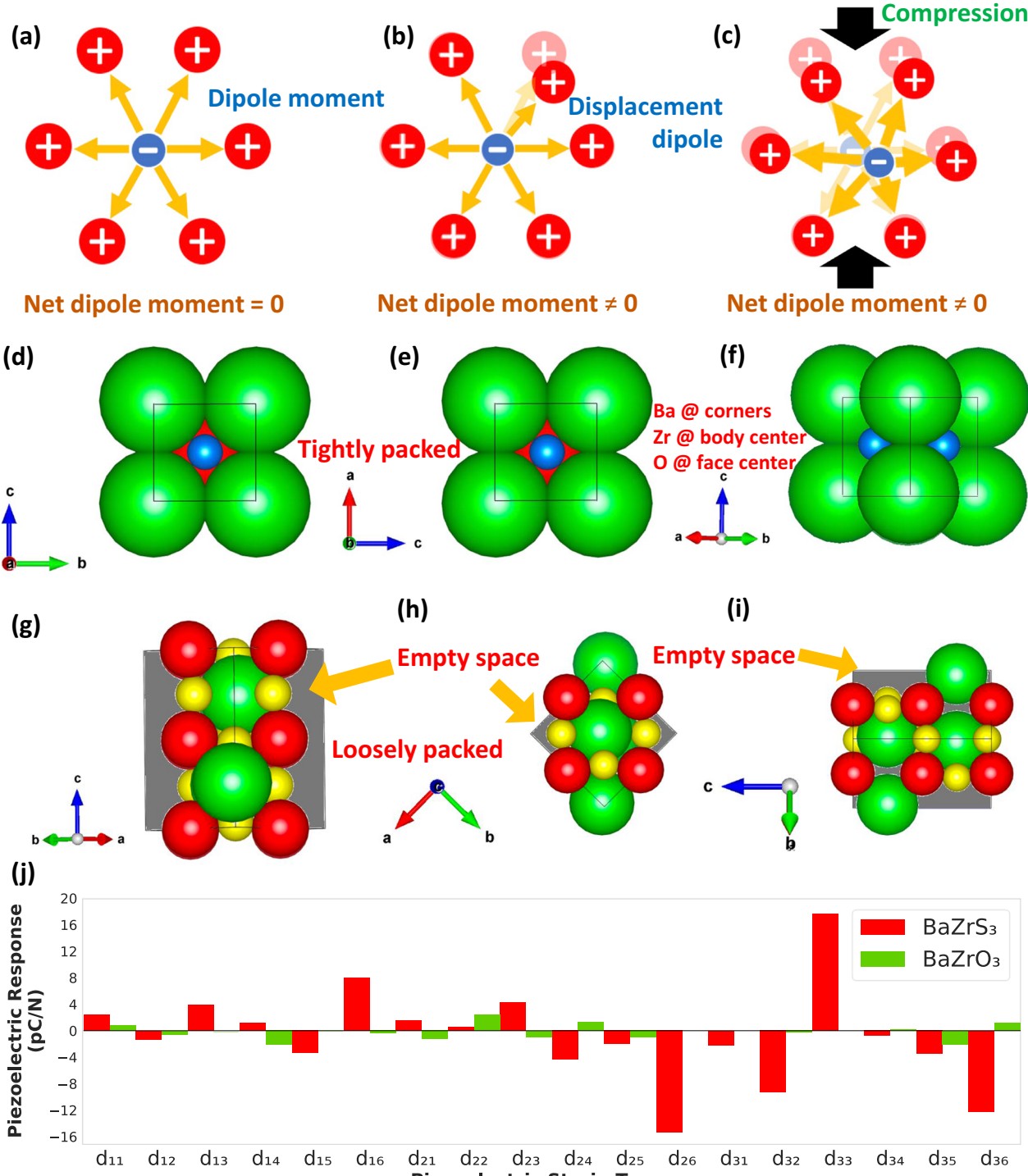

**Fig. 2 | Dipole moment in ionic systems, a structural comparison between cubic BaZrO₃ and orthorhombic BaZrS₃, and first-principles calculation of piezo-electric coefficients. a** A schematic representation of a net zero dipole moment, where an anion is surrounded by cations. **b** The transition to a non-zero dipole moment when an ion is displaced from its equilibrium position. **c** Symmetry breaking under load can produce a quantifiable displacement-induced dipole moment, even in centrosymmetric crystals. **d**, **e** Top and side views of a cubic BaZrO₃ (Pm3̄m) unit cell consisting of 5 atoms, illustrating Zirconium (Zr) at the body center (in red), Oxygen (O) at the face center (in blue) and Barium (Ba) at the corners in green. **f** The 3D isometric representation of BaZrO₃ reveals a densely packed unit cell, indicating minimal space for ion displacement. **g**–**i** The different projections of the orthorhombic BaZrS₃ (***Pnma***) unit cell that is comprised of 20 atoms and contains a significant volume of vacant space; yellow represents Sulfur (S). The loosely packed structure of BaZrS₃ allows for reducing symmetry via extended displacement of the ions, resulting in an enhanced dipole moment. **j** Comparison of the DFPT calculated piezoelectric strain tensors (d$_{ij}$) for BaZrS₃ and BaZrO₃. The piezoelectric tensors were obtained by averaging from an ensemble of 20 distinct randomly generated structural variations (see Methods).

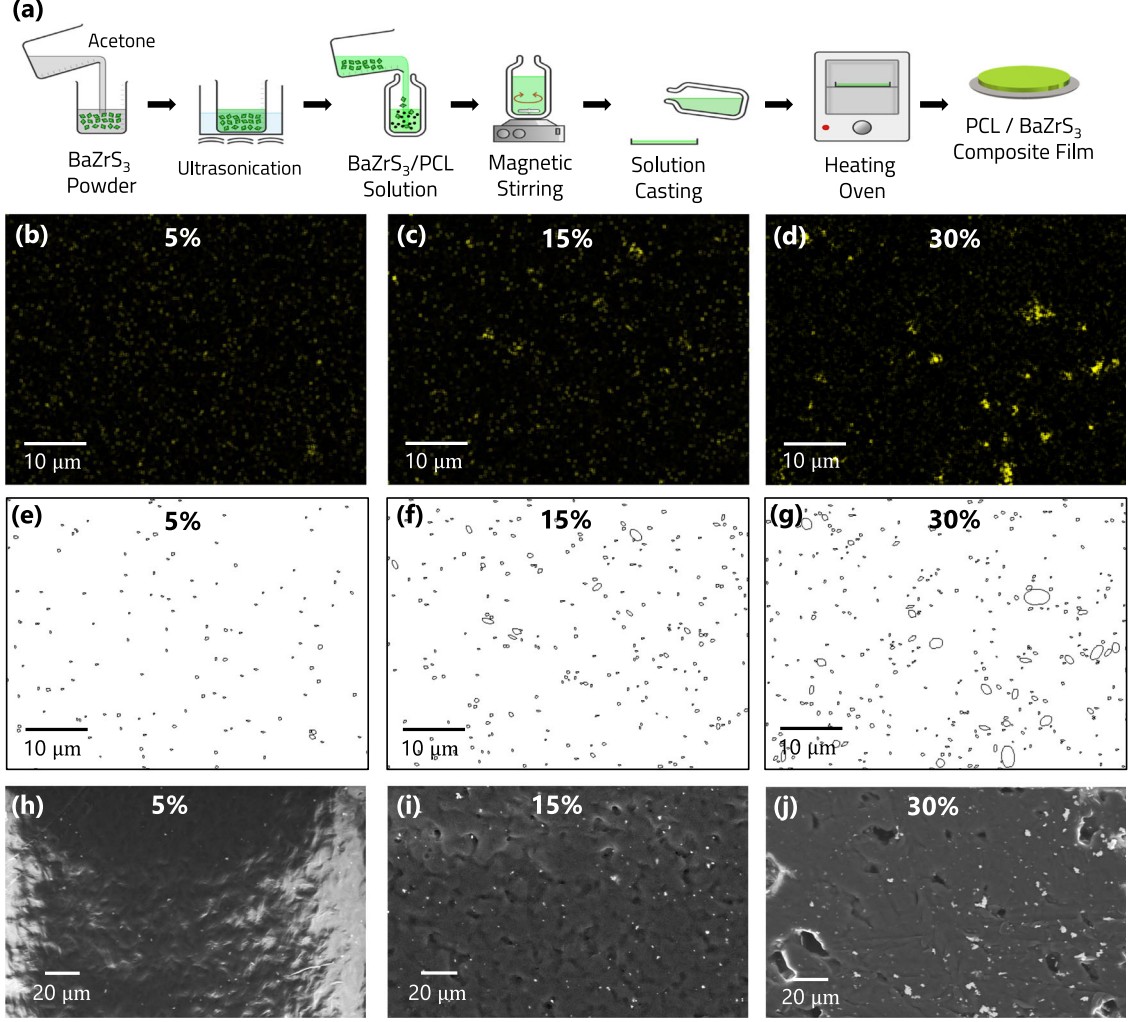

**Fig. 3 | BaZrS₃ encapsulated PCL film synthesis and structural characterization.** **a** Schematic illustration of the solvent casting process used to disperse BaZrS₃ particles in PCL. **b–d** EDS elemental maps of sulfur in PCL-BaZrS₃ composite films at 5, 15 and 30 wt.% loading respectively. **e–g** Particle size analysis of elemental maps **b–d**. **h–j** Backscattered electron detector (BSD) images from SEM of PCL-BaZrS₃ composite films at 5, 15 and 30 wt.% loading respectively.

typical backscattered electron microscopy cross section image of the PCL/BaZrS₃ composite film at ~10 wt% BaZrS₃ loading fraction is provided in Supplementary Fig. S9 showing the PCL/BaZrS₃ composite film (~0.35 mm thick) sandwiched between the Cu electrodes. A pneumatic actuator was used to control the applied pressure as well as the frequency of the applied load. Figure 4b, e shows the open circuit voltage (V_oc) and short circuit current (I_sc) as a function of time, at a fixed applied pressure of ~30 pounds inch⁻² (PSI) and ~4 Hz load frequency. Data is shown for the baseline PCL film (which is weakly piezoelectric) as well as composite films at six different BaZrS₃ loading fractions (from ~2 wt% up to ~30 wt%). We find that there is an optimum BaZrS₃ loading fraction of ~15 wt%. At this optimum loading, the V_oc and I_sc values for the composite material were ~4-fold and ~5.3-fold higher, respectively, than the pure PCL material. The reason behind loss of piezoelectric performance at high loadings (>15 wt%) is evident in the back-scattered electron detector images in Fig. 3h–j. At high loading fractions, we find that there is significant agglomeration of BaZrS₃ particles which leads to a weak particle-matrix interface. This leads to particle debonding– such debonded particles are electrically and mechanically disconnected from the polymer network and hence do not contribute to the piezo-response.

The piezoelectric response of the PCL-BaZrS₃ composite film is observed to be highly stable over extended pressure cycling as indicated in Fig. 4f, g. As is to be expected, the piezo-response of the PCL-

BaZrS₃ material improves with increasing applied pressure (Fig. 4c) and loading frequency (Fig. 4d). Finally, the PCL-BaZrS₃ film (~30 PSI and ~4 Hz input; BaZrS₃ loading of ~15 wt%) was tested against various resistive loads to establish the area-normalized power output (Fig. 4h) —a peak output of ~103.5 µW cm⁻² was measured. In Fig. 4i, we compare the power harvesting ability of the BaZrS₃-PCL film with the available literature[26–31] for PCL composites with various state-of-the-art piezoelectric additives. Clearly, the BaZrS₃-PCL material's performance is significantly better than previous literature reports, which indicates the potential of this class of material for mechanical energy harvesting. We also compared the V_oc and I_sc response (at ~30 PSI pressure and ~4 Hz load frequency) for composite films with BaZrO₃ and BaZrS₃ additives at the same loading fraction of ~15 wt%. The results shown in Supplementary Fig. S10 indicate that BaZrS₃ additives offer much improved piezoelectric performance relative to BaZrO₃, which is consistent with the DFPT calculations shown in Fig. 2.

We have varied the thickness of the PCL/BaZrS₃ composite film in the 0.2 to 1 mm range. For all thickness values, the BaZrS₃ loading fraction was held constant at ~10 wt%. The results indicate that the open circuit voltage improves with film thickness (Supplementary Fig. S11). This is due to the increased total amount of piezo-material in the thicker films. However, this improvement tends to saturate for thicker films, since the electrodes on either side of the piezoelectric film are unable to collect charges efficiently as the films get thicker[32].

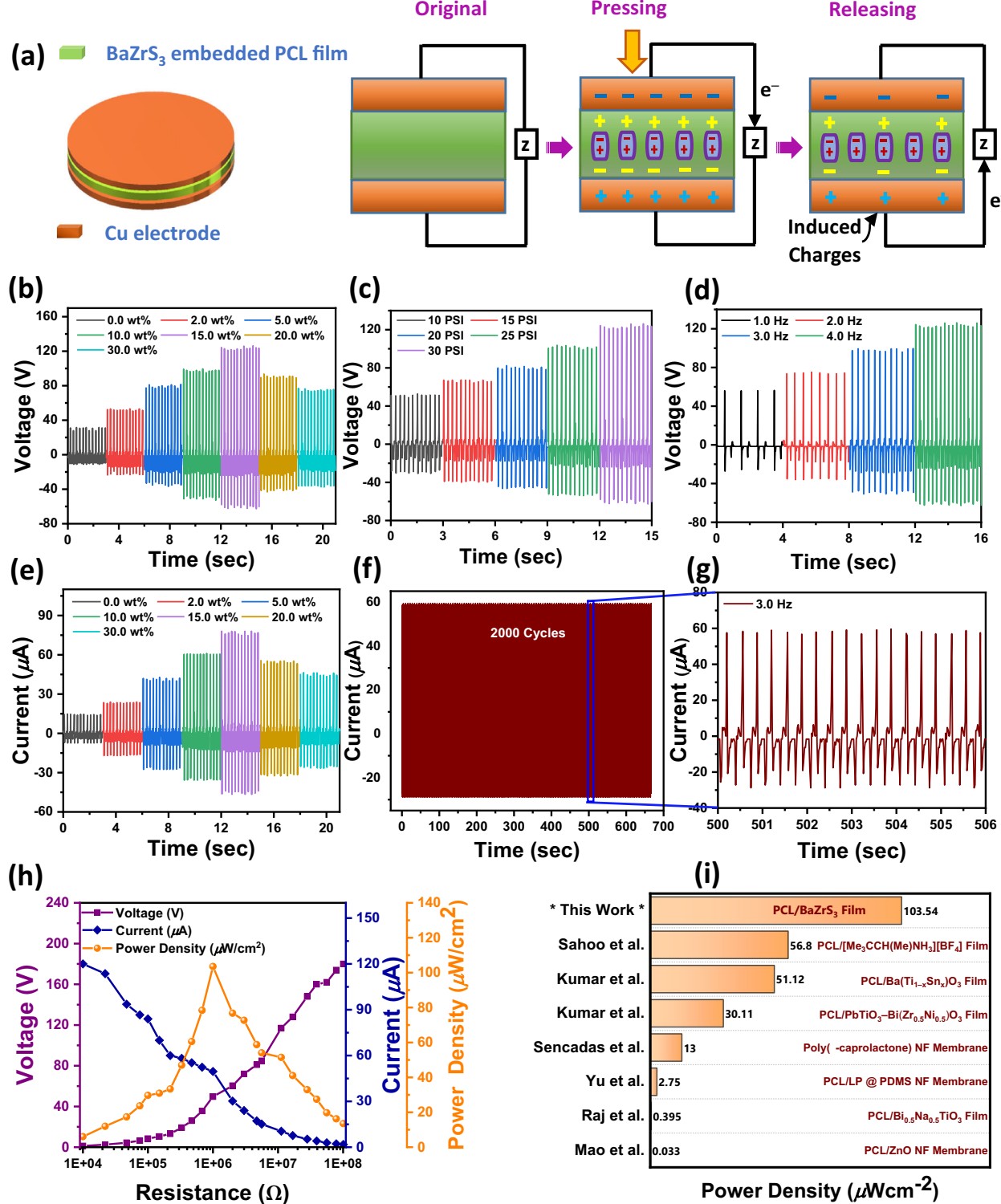

**Fig. 4 | Piezoelectric characterization of the BaZrS$_3$-PCL composite material.**
**a** Device structure and loading configuration. **b** Open circuit voltage (V$_{oc}$) and
**e** short circuit current (I$_{sc}$) at -30 PSI and -4 Hz load frequency for different BaZrS$_3$
weight (wt) fractions. V$_{oc}$ under **c** various loading pressures (at -4 Hz) and
**d** different loading frequencies (at -30 PSI) for -15% BaZrS$_3$ weight fraction. Stability
and durability test **f** under 2000 working cycles and **g** enlarged view of the
response for testing at -30 PSI, -3.0 Hz load frequency and -15% BaZrS$_3$ weight
fraction. **h** Average areal power density for various resistive loads (pressure: -30
PSI; load frequency: -4.0 Hz; BaZrS$_3$ weight fraction: -15%). **i** Areal power density
comparison of BaZrS$_3$/PCL with the available literature[26–31] for PCL films.

Another contributory factor is the polarization saturation effect[33]
which gets more pronounced with increasing film thickness. Besides
PCL, we also performed experiments with Polymethyl Methacrylate
(PMMA) as the polymer matrix. Open circuit voltage (V$_{oc}$) vs. time plots

are shown in Supplementary Fig. S12 for -0.35 mm thick PMMA films in
the 1.0–4.0 Hz frequency range at 30 PSI pressure. The test data indi-
cates that PMMA shows negligible piezoelectric response. Under
identical test conditions the PMMA/BaZrS$_3$ composite film (-10 wt%

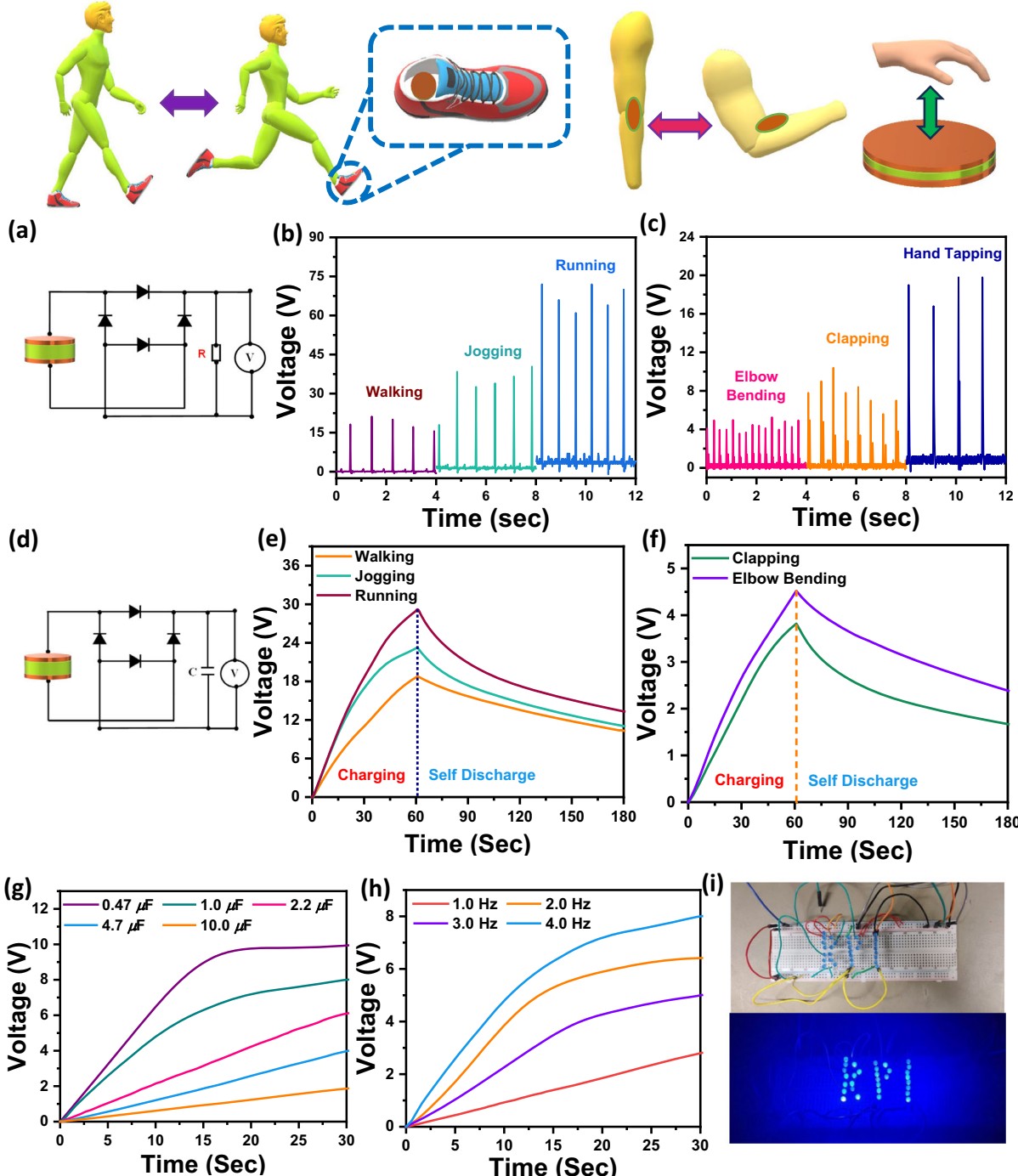

**Fig. 5 | Harvesting energy from body movement using the 15 wt% PCL-BaZrS₃ composite device. a** Rectified $V_{oc}$ circuit output connected with 1MΩ resistance for harvesting energy from a variety of body movements. **b** Regular walking, jogging, and running; **c** elbow bending, clapping, and hand tapping voltage output. **d** Capacitor test circuit. Charging of the device (for 60 s) and self-discharge (for 120 s) during **e** regular walking, jogging, and running; **f** hand clapping and elbow bending with 1.0 μF capacitor. Device charging ability by hand tapping for 30 s with **g** different capacitors at 4 Hz tapping frequency, and **h** charging with 1.0 μF capacitor at various hand tapping frequencies. **i** LED illumination proof-of-concept demonstration.

BaZrS₃ loading), exhibits a pronounced maximum $V_{oc}$ of ~62.4 V, confirming that the generated response is predominantly from BaZr₃ and not from the polymer.

Finally, we demonstrate practical mechanical (vibrational) energy harvesting applications of the BaZrS₃-PCL material. To accomplish this, composite films with ~15 wt% BaZrS₃ loading were tested in conjunction with a full bridge rectifier and 1.0 MΩ load resistance (Fig. 5a); this resistance value gives the optimal power density (see Fig. 4h). The resulting direct current (DC) voltage obtained for regular walking,

jogging, and running is indicated in Fig. 5b. Similarly elbow bending, routine clapping, and hand tapping at ~1.0 Hz frequency also generates a DC voltage as indicated in Fig. 5c. For regular walking, jogging, and running, the resulting maximum DC voltage was ~21.2, ~40.4, and ~72 V, while it was ~5.3, ~10.4, and ~19.8 V for elbow bending, clapping, and hand tapping, respectively.

Next, we demonstrated the ability of the BaZrS₃-PCL vibrational energy harvester to charge various small electronic and electrochemical devices. For this, we added an ~1.0 μF capacitor to the fully

rectified circuit (Fig. 5d) and repeated the aforementioned body movement modes for ~60 seconds (i.e., charging) and then monitored capacitor self-discharge over the next ~120 s. It was observed that the capacitor can charge up to ~18.8, ~23.2, and ~29.2 V for walking, jogging, and running, respectively (Fig. 5e), whereas it could charge to a much reduced ~3.82 and ~4.52 V for elbow bending, and clapping, respectively (Fig. 5f). A range of capacitors with capacitance up to ~10.0 μF were successfully charged by hand tapping at ~4 Hz (Fig. 5g), with the charging rate gradually decreasing with increasing capacitance. The device was further tested with a ~1.0 μF capacitor under various tapping frequencies, indicating that higher frequencies significantly accelerate the charging process (Fig. 5h). Finally, the BaZrS$_3$-PCL device was connected to a series of commercial LEDs via a full-wave bridge rectifier and a ~0.1 μF capacitor; the device was able to successfully illuminate 35 LEDs (each requiring 3.0–3.2 V) (Fig. 5i) simply by hand tapping at ~4 Hz. Besides its primary energy harvesting function, we find that the BaZrS$_3$-PCL device connected to the bicep, forearm, quadricep and calf muscle, can also serve as a self-charging biomechanical movement sensor (Supplementary Fig. S13).

It should be noted that flexoelectricity[34] can induce polarization even in centrosymmetric materials, hence it is important to ascertain the extent to which flexoelectricity plays a role for our specific material system. To study this, we performed detailed ab initio calculations to study the role of flexoelectricity in BaZrS$_3$. The results (Supplementary Figs. S14, S15) indicate that the role of flexoelectricity in generation of polarization or non-zero dipole moment under deformation is minimal for BaZrS$_3$. Another independent evidence that the flexoelectric effect is negligible in BaZrS$_3$ comes from the testing of the PCL/BaZrS$_3$ composite films. Flexoelectricity requires a large strain gradient to polarize the material[34,35]. However, in Fig. 4, a uniform pressure load is applied along the thickness direction resulting in a uniform compression of the material. Even though there are no large strain gradients, a pronounced power generation is reported in our material. The test data in Fig. 4i indicates that the BaZrS$_3$ additive outperforms other well-known piezoelectric material additives, which builds confidence that this is a piezoelectric and not a flexoelectric effect. The absence of two different triboelectric layers and the absence of separation or gap between the Cu tape and the PCL/BaZrS$_3$ composite film also rules out the possibility of triboelectric effects[36,37].

To summarize, we report piezoelectricity in the family of chalcogenide perovskite materials. This was established both experimentally (by direct PFM measurements on BaZrS$_3$ and the testing of BaZrS$_3$-polymer films) and theoretically (by first-principles DFPT calculations). The first-principles calculations indicate that the loosely packed unit cell of chalcogenide perovskites with significant volume of vacant space, enables symmetry reduction via an extended displacement of the ions, resulting in an enhanced dipole moment, and pronounced piezo-response. To demonstrate practical applications, we dispersed BaZrS$_3$ particles in polymers, and built devices to harvest vibrational energy from ubiquitous body movements for charging capacitors and powering light emitting diodes. The chalcogenide perovskites reported here represent piezoelectric materials that are completely lead-free and comprised of environmentally benign, non-toxic and relatively earth-abundant materials. It should be noted that defects are unavoidable in any material system and such defects could in principle further break the symmetry and further amplify the piezoelectric response. In fact, defect-engineering could be an avenue to intentionally break[38–40] the centrosymmetry of chalcogenide perovskites and thus further amplify the piezoelectric response in this class of material and should be pursued as part of future research.

## Methods
### Computational methods
All first-principles calculations were performed using the Vienna ab-initio Simulation Package (VASP)[41]. Electronic configurations [Xe] 6s$^2$,

[He] 2 s$^2$ 2p$^4$, [Ne] 3 s$^2$ 3p$^4$ and [Kr] 4d$^2$ 5 s$^2$ were adopted to represent Ba, O, S and Zr, respectively. The Projector Augmented Wave (PAW) method was employed to describe the atomic interactions within the system[42]. The Perdew–Burke–Ernzerhof (PBE) generalized gradient approximation (GGA) functional was chosen to reproduce the exchange-correlations[43]. The planewaves expansion was considered with an energy cutoff of 800 eV. For the Brillouin zone, a k-points mesh grid was generated using the Monkhorst–Pack method with a precision level of 0.02[44]. To ensure convergence of the Kohn-Sham equations through the self-consistent field procedure, a threshold of $10^{-6}$ eV was established as the convergence criterion between successive iterations. The evaluation of the piezoelectric response in centrosymmetric compounds is not feasible within their relaxed ground state unless the inherent symmetry is perturbed. To disrupt the symmetry, we introduce randomly generated atomic displacements (1-2% from their respective coordinates) for all ions within the unit cell, and subsequently compute the piezoelectric tensors by averaging from an ensemble of 20 distinct structural variations. The input files and coordinate of structure files are uploaded in git-hub (https://github.com/pshatom/piezo-files); this information can be used to reproduce results.

### Synthesis of BaZrS$_3$ powder
1 g BaZrO$_3$ (powder, <10 μm particle size from Sigma-Aldrich) was kept in a quartz boat and sulfurization was performed in MTI OTF-1200X three-zone tube furnace (quartz tube with 3 inches diameter) using CS$_2$ as the sulfur source at a temperature of ~1050 °C for ~4 h. A mass flow controller was used to keep the flow rate of the CS$_2$–N$_2$ mixture at ~5 sccm, while a pressure of ~2 Torr was maintained inside the tube. The tube was purged with ultrahigh purity N$_2$ until the furnace was cooled down completely. The tube was then brought to atmospheric pressure to take out the sulfurized powder.

### Synthesis of BaZrS$_3$ film
~1.92 g of barium acetate (99%, AlfaAesar), ~3.66 g of zirconium(IV) acetylacetonate (97%, Sigma-Aldrich), and ~0.90 g of polyvinyl butyral (Sigma-Aldrich) were stirred and dissolved in ~25 ml of propionic acid (99.5%, Sigma-Aldrich) at 60 °C to obtain the precursor solution for BaZrO$_3$ thin film. The clear solution was spin-coated on a clean quartz substrate (1 cm × 1 cm × 2 mm) at ~2000 rpm for ~1 min followed by ~5000 rpm for ~5 min. The spin-coated film was annealed in the air in a Thermolyne FB1315M muffle furnace at ~700 °C for~ 90 min followed by ~40 min at ~870 °C to obtain BaZrO$_3$ thin films. These were then sulfurized similarly to the BaZrO$_3$ powder using MTI OTF-1200X three-zone tube furnace with CS$_2$ as the sulfur source at a temperature of ~1050 °C for ~4 h to obtain BaZrS$_3$ thin films[10].

### Preparation of BaZrS$_3$ encapsulated PCL film
Initially, the BaZrS$_3$ particles were added in ~14 g acetone solvent (Reagent grade acetone ≥99.5% ACS, VWR International) and then sonicated using Fabulustre Ultrasonic Cleaners for ~20 min in a water bath to get uniform dispersion of the particles. To prepare BaZrS$_3$ incorporated PCL solution, PCL (Polycaprolactone with Mw = 80,000, Sigma Aldrich, St.Louis, MO, USA) was added in the acetone in a ~20 mL glass vial to make polymer solutions. The solution was magnetically stirred at ~60 °C and ~200 rpm for ~12 h to ensure complete dissolution of PCL in acetone. The homogeneous solution mixture was then cast on a petri dish and kept in the oven for ~4 h at ~45 °C to obtain an acetone-free solid and flexible PCL/BaZrS$_3$ composite film. The concentration of BaZrS$_3$ in the solution was varied relative to PCL to fabricate 0.0 (control), 2.0, 5.0, 10.0, 15.0, 20.0, and 30.0 wt% concentrated PCL/BaZrS$_3$ composite film.

### Fabrication of PCL-BaZrS$_3$ piezoelectric device
The PCL/BaZrS$_3$ composite film was removed from the petri dish and cut into circular shapes with a diameter of ~5.5 cm and a thickness of

-0.35 ± 0.03 mm. Then commercially purchased conductive copper tape was attached on both sides of the composite film, to serve as electrodes. All electrical testing of the fabricated device was conducted in ambient conditions.

## Materials characterization

XRD results were obtained through a PANalytical X'Pert Pro diffractometer using Cu Kα (λ = 1.5405 Å) radiation with an X-ray generator at 45 kV and 40 mA. Scanning electron microscopy (SEM) was done on Supra 55 FESEM at 2.5 kV.

## Piezoresponse Force Microcopy (PFM)

PFM was done on Asylum MFP3D atomic force microscopy (AFM) system (https://mmrc.caltech.edu/Asylum/Manuals/AR_Applications Guide_16A.pdf) using ASYELEC.01-R2 conductive probes which had a coating of Ti/Ir, tip radius of ~25 nm, spring constant of ~2.8 N/m and a free air resonance frequency of ~75 kHz. A high-stiffness cantilever was chosen to minimize the influence of electrostatic interactions on the piezoelectric measurements. The typical contact resonance frequency of this conductive tip is ~340 kHz. The frequency of the signal applied to the sample was close to this resonance peak, which is significantly higher than the low-pass cutoff frequency of the AFM topography feedback loop. A consistent tip pressure helped reduce variability in measurements caused by topographical variations. PFM calibration was performed using the GetRealTM calibration procedure outlined in the application guide for the Asylum research MFP-3D system (https://mmrc.caltech.edu/Asylum/Manuals/AR_ApplicationsGuide_16A.pdf).

The PFM setup with the BaZrS$_3$ film on quartz is shown in Supplementary Fig. S16. In this set-up, a conductive AFM tip serves both as the top electrode and the sensing element. The main limitation of this set-up is that the field generated beneath the AFM tip is highly concentrated due to the nanoscale size of the tip, which can result in inaccuracies[45]. This problem can be mitigated[46–48] by coating the top surface of the piezoelectric film with a thin conductive coating to reduce charge concentration. This method of top coating the piezoelectric film also generates a relatively uniform electric field and surface deformation, making it preferable for accurately detecting the piezoelectric response[49–51]. We therefore sputter coated our BaZrS$_3$ films with a ~5 nm layer of Pt. Given the thinness of Pt relative to the BaZrS$_3$ film (~300 nm), its impact on the PFM results are considered negligible[46–51]. To prove that our set-up can discriminate between materials, we also carried out PFM characterization for BaZrO$_3$ films on quartz substrate with a thin ~5 nm layer of Pt on the top surface. BaZrO$_3$ was chosen since ab initio calculations indicated that BaZrO$_3$ exhibits weak piezoelectricity relative to BaZrS$_3$ (Fig. 2). The test results (Supplementary Figs. S17, 18) confirmed a minimal change in piezoresponse amplitude with increasing voltage for BaZrO$_3$, indicating that the PFM setup was capable of detecting and differentiating between the piezoelectric responses of different materials.

## Electrical measurement

The open circuit voltage of the composite piezoelectric devices was measured by Agilent Technologies DSO1014A Oscilloscope, while short circuit current was measured by an Edmund Optics current amplifier connected to the oscilloscope. During the voltage and current measurement, pressure was applied by an NF10A-E02-AMCM0 Norgren NFPA cylindrical pneumatic actuator. The rectified voltage was measured by connecting the device with a full bridge rectifier. The capacitance charging and discharging data were obtained by connecting the full bridge rectified device to a Keithley 4200A-SCS Parameter Analyzer.

## Data availability

All data generated in this study have been deposited in the following public repository [github−pshatom/piezo-files] which can be accessed at https://github.com/pshatom/piezo-files/tree/main/Raw%20data%20figures.

## Code availability

All code and associated input and structure files generated in this study have been deposited in the following public repository [github−pshatom/piezo-files] which can be accessed at https://github.com/pshatom/piezo-files/.

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

## Acknowledgements

N.K. acknowledges funding support from the USA National Science Foundation (award number 2013640) and the John A. Clark and Edward T. Crossan chair professorship at the Rensselaer Polytechnic Institute. J.S acknowledges funding support from the USA National Science Foundation (award number 2236099). G.B. acknowledges funding support from the USA National Science Foundation (award number 1944040).

## Author contributions

S.S. synthesized the $BaZrS_3$ materials, and performed material-level characterization, S.S.H.A. synthesized polymer composites with $BaZrS_3$ additives and carried out the energy harvesting characterization work, P.S. and G.B. carried out first-principles calculations to determine piezoelectric coefficients, S.S. and S.K. carried out the piezoelectric force microscopy characterization, J.S. and N.K. conceived and directed the project. S.S., S.S.H.A., P.S., G.B., J.S. and N.K. wrote the manuscript.

## Competing interests

The authors declare no competing interests.
