## [Peer Review File · Nature Communications]

REVIEWER COMMENTS

Reviewer #1 (Remarks to the Author):

Abir et al. report piezoelectricity in lead-free chalcogenide perovskite BaZrS₃ using piezoresponse force microscopy measurements. Through first-principles calculations, despite BaZrS₃'s centrosymmetric structure, it's argued that such materials readily polarize under deformation due to a loosely packed unit cell with significant vacant space. This unique structure, calculations show, allows for extended ion displacement, leading to symmetry reduction and an enhanced displacement-mediated dipole moment. Under sinusoidal pressure loading, composites of BaZrS₃ particles dispersed in a polycaprolactone (PCL) matrix are shown to generate a peak areal power density of $\sim 103.5 \mu\text{W cm}^{-2}$ and are shown to harvest energy from routine body movements. The study is interesting in a highly topical area. However, there are several issues, and conclusions in their present form are not supported by experimental data.

Detailed comments concerning the manuscript are:

1. What is the thickness of the BaZrS₃ films used for the PFM study? It's not mentioned in the manuscript.
2. PFM studies are performed on thin films of BaZrS₃ fabricated on quartz substrates with any conductive bottom electrode. This is not a standard configuration for PFM, so, how do authors interpret their response which should be different compared to the standard configuration?
3. PFM is used as the sole evidence for the piezoelectricity of the BaZrS₃ films. Obtained PFM images (Fig.1) show quite a non-uniform piezoresponse, which can arise from multiple extrinsic factors (e.g. morphological channel crosstalk, electrostatic contributions, and field-induced ion dynamics) and not just the intrinsic piezoelectricity. How do authors rule out these various extrinsic contributions and conclude that the experimental results support the suggested mechanism?
4. Theoretical calculations show centrosymmetric BaZrS₃ is polarized under deformation due to a loosely packed unit cell with significant vacant space. In PFM measurements, however, an electric field is applied and an apparent strain response is detected. So the question arises, do theoretical calculations support electrically driven ionic displacement leading to microscopic strain that is large enough to detect?
5. How is the magnitude of the piezoelectricity (e.g. d_{33}) expected to scale with increasing deformation?

6. What's the role of flexoelectricity if any in the appearance of non-zero dipole moment under deformation? This is given flexoelectricity can induce polarization even in centrosymmetric materials as has been evidenced by many reports in literature.

7. How is the calibration done of the PFM response presented in Fig 1h? PFM images Fig1c-g have arbitrary units, however, Fig. 1h shows units of pm for piezoresponse displacement. Moreover, the measurements are performed at resonance, which should be deconvoluted to get the true intrinsic response. The authors should provide more information on this.

8. For practical energy harvesting and power density demonstration, BaZrS3 powder is mixed with a polymer PCL, which is a weakly piezoelectric material. It would be ideal to conduct these experiments with a non-piezoelectric polymer in order to ensure that the generated response is more or less coming entirely from BaZrS3 and not from the polymer. From the presented data, it is difficult to draw a definitive conclusion.

9. Why do the Voc and Isc increases up to ~15% BaZrS3 weight fraction, and decrease with a further increase in the weight fraction of BaZrS3? Significant agglomeration/voids seem to happen at ~30% BaZrS3 weight fraction.

10. What role does triboelectricity play in the generated response, if not much, how do authors rule that out?

Reviewer #2 (Remarks to the Author):

In this manuscript, the authors provided First-principles calculations, piezoresponse force microscopy, and BaZrS3/PCL device characterization to demonstrate the unique BaZrS3 material to piezoelectric applications. The highly improved piezoresponse of BaZrS3 compared to BaZrO3 shows the potential of such material in the energy generation of lead-free chalcogenide perovskite material-based wearable electronics. The material is novel and the topic is interesting. Therefore, I recommend this manuscript to be accepted after addressing the following comments in the minorly revised manuscript.

1. In Figure 1 c-g and Figure S4, please provide the height sensor images corresponding to the same area of amplitude and phase images.

2. For peak of XRD pattern, please note the corresponding crystal plane.

4. Please provide the hysteresis loop, amplitude-voltage curve, and phase-voltage curve to prove the ferroelectricity properties of BaZrS₃.

5. Please provide the cross-section SEM images of the device and note the thickness of individual layers. Would the thickness of the BaZrS₃/PCL affect the overall piezoelectric behavior of the device?

6. Please cite the following papers on BaZrS₃ and piezoelectric perovskites to attract broad interest.

<https://doi.org/10.1002/anie.202301049>

<https://doi.org/10.1016/j.jallcom.2023.170457>

<https://doi.org/10.1016/j.nanoen.2019.104317>

<https://doi.org/10.1002/adv.202104703>

<https://doi.org/10.1016/j.cej.2021.133620>

<https://doi.org/10.1002/adv.202105974>

Reviewer #3 (Remarks to the Author):

The authors find that BaZrS₃, a material with a centrosymmetric structure, shows an unusually large piezoelectric response. The manuscript is well-written, and the experiments mostly support the conclusion. The findings are interesting and will attract a broad audience. I can recommend the publication of this work if the authors address my following concerns:

1. The paper only reported d_{33} . I'm wondering if the authors measured other components of the piezoelectric strain tensor. A related concern is: since the sample is polycrystalline, it's difficult to guarantee the applied pressure is along the z-direction. Then how can the authors decide if they measured d_{33} , instead of d_{22} or d_{11} ?

2. In the theoretical analysis on the origin of the piezoelectricity, the authors performed DFPT calculations for 20 randomly distorted structures (1-2% from their respective coordinates). In my experience, the distorted structures (distorted by 1-2%) typically have imaginary phonons, which can hinder the following calculations. How does the authors overcome this issue? Also, are the calculated results dependent on the generated distorted structures? How large are the supercells used for the distorted structures? To help reproducing the calculation, I suggest the authors provide key input/output files of the calculations.

3. Is the method (computing piezoelectric tensors using the average of some randomly distorted structures) used by previous work? If yes, please provide reference.

4. On the origin of piezoelectricity, can the author discuss the possibility of defect-induced piezoelectricity in BaZrS₃? How do they rule out this possibility?

Decision: reconsider for publication following response to the questions.

RESPONSE LETTER

[Reviewer #1's Comments]: *In this manuscript, the authors provided first-principles calculations, piezoresponse force microscopy, and BaZrS₃/PCL device characterization to demonstrate the unique BaZrS₃ material to piezoelectric applications. The highly improved piezoresponse of BaZrS₃ compared to BaZrO₃ shows the potential of such material in the energy generation of lead-free chalcogenide perovskite material-based wearable electronics. The material is novel and the topic is interesting. Therefore, I recommend this manuscript to be accepted after addressing the following comments in the minorly revised manuscript.*

Thank you for your support and encouragement.

We have tried our best to address your specific comments as listed below and have made changes accordingly to the revised manuscript. All changes in the main manuscript have been highlighted in red color for your kind attention.

[Comment (1)] *In Figure 1 c-g and Figure S4, please provide the height sensor images corresponding to the same area of amplitude and phase images.*

Thank you for your advice. We have added height sensor images to Figure S2 and S4. Figure S2 corresponds to the standard PPLN sample and Figure S4 corresponds to the BaZrS₃ sample. Figure 1c-g are the same as Figure S4 so they were not updated.

[Comment (2)] *For peak of XRD pattern, please note the corresponding crystal plane.*

The corresponding crystal planes for the various peaks in the XRD pattern have been added to Figure S1b (shown below for your reference).

Fig. R1 (Fig. S1b in Supplementary Information): XRD and Rietveld refined profile of BaZrS₃ film over quartz substrate.

[Comment (3)] Please provide the hysteresis loop, amplitude-voltage curve, and phase-voltage curve to prove the ferroelectricity properties of BaZrS₃.

Our current Piezoresponse Force Microscopy (PFM) system, the Asylum Research MFP-3D, presents certain technical limitations that restrict our ability to provide these measurements. Specifically, this AFM system is capable of applying a maximum voltage of only 10 volts. This voltage limit is insufficient to achieve the necessary electric field strength required to induce and measure a hysteresis loop in BaZrS₃.

[Comment (4)] Please provide the cross-section SEM images of the device and note the thickness of individual layers. Would the thickness of the BaZrS₃/PCL affect the overall piezoelectric behavior of the device?

The backscattered electron microscopy cross section image of the PCL/BaZrS₃ composite film at ~10 wt% BaZrS₃ loading fraction is provided below (**Fig. R2**) and has also been included in the revised supporting information file.

Fig. R2 indicates that the copper (Cu) electrode thickness is ~0.015 mm and the PCL/BaZrS₃ composite film thickness is about 0.35 ± 0.03 mm.

Fig. R2 (also included in Supplementary Information): Backscattered electron microscopy cross-section images of a PCL/BaZrS₃ composite film sandwiched between two Cu tapes that serve as electrodes. The BaZrS₃ loading fraction in this case is ~10 wt%.

Following your advice, we systematically varied the thickness of the PCL/BaZrS₃ composite film in the 0.2 to 1 mm range. For all thickness values, the BaZrS₃ loading fraction was held constant at ~10 wt%. The results indicate that the open circuit voltage improves with film thickness (**Fig. R3**, also included in the revised supporting information file). This is due to the increased total amount of piezo-material in the thicker films. However, this improvement tends to saturate for thicker films (**Fig. R3**), since the electrodes on either side of the piezoelectric film are unable to collect charges efficiently as the films get thicker [1]. Another contributory factor is the polarization saturation effect [2] which gets more pronounced with increasing film thickness.

Fig. R3 (also included in Supplementary Information): Open circuit voltage response of PCL/BaZrS₃ (~10 wt%) composite film at ~0.20, ~0.35, ~0.70, and ~1.0 mm film thickness.

References cited in Response to Comment (4) of Reviewer 1:

1. O. F. Ünsal, Y. Altın & A. Ç. Bedeloğlu, Flexible Electrospun PVDF Piezoelectric Nanogenerators with Electro-spray-Deposited Graphene Electrodes. *J. Electron. Mater.* 52, 2053-206 (2023).
2. X. Du et al. Porous, multi-layered piezoelectric composites based on highly oriented PZT/PVDF electrospinning fibers for high-performance piezoelectric nanogenerators. *J. Adv. Ceram.* 11, 331-344 (2022)

[Comment (5)] Please cite the following papers on BaZrS₃ and piezoelectric perovskites to attract broad interest.

<https://doi.org/10.1002/anie.202301049>

<https://doi.org/10.1016/j.jallcom.2023.170457>

<https://doi.org/10.1016/j.nanoen.2019.104317>

<https://doi.org/10.1002/advs.202104703>

<https://doi.org/10.1016/j.cej.2021.133620>

<https://doi.org/10.1002/advs.202105974>

Thank you for suggesting these excellent papers. They have all been cited in the revised submission.

[Reviewer #2's Comments] *The authors find that BaZrS₃, a material with a centrosymmetric structure, shows an unusually large piezoelectric response. The manuscript is well-written, and experiments mostly support the conclusion. The findings are interesting and will attract a broad audience. I can recommend publication of this work if authors address my following concerns:*

Many thanks for your support, encouragement and helpful advice.

We are grateful to you for your time and effort in helping us improve the manuscript. Our point-by-point response to your suggestions is provided below. All changes made in the revised manuscript are highlighted in red color.

[Comment (1)] *The paper only reported d_{33} . I'm wondering if the authors measured other components of the piezoelectric strain tensor. A related concern is: since the sample is polycrystalline, it's difficult to guarantee the applied pressure is along the z-direction. Then how can the authors decide if they measured d_{33} , instead of d_{22} or d_{11} ?*

Please note that the piezoresponse force microscopy (PFM) characterization method measures the effective piezoelectric charge coefficient (i.e., $d_{33, effective}$). For a randomly oriented polycrystalline material (such as BaZrS₃), $d_{33, effective}$ represents the overall piezoelectric response in the direction normal to the sample surface, which is also the direction of the applied electric field. Hence in Fig. 1 of the main paper, we report $d_{33, effective}$ for our material. This has now been clarified in the text of the revised manuscript.

[Comment (2)] *In the theoretical analysis on the origin of the piezoelectricity, the authors performed DFPT calculations for 20 randomly distorted structures (1-2% from their respective coordinates). In my experience, the distorted structures (distorted by 1-2%) typically have imaginary phonons, which can hinder the following calculations. How does the authors overcome this issue? Also, are the calculated results dependent on the generated distorted structures? How large are the supercells used for the distorted structures? To help reproducing the calculation, I suggest the authors provide key input/output files of the calculations.*

Thank you for raising these important points.

In addressing the potential issue of imaginary phonons in distorted structures, we have rigorously validated our results by demonstrating the absence of imaginary phonon modes in the phonon band structure of our specific material (Fig. R4, also included in the revised supporting information). The calculated band structures exhibit minimal sensitivity to the specific distorted structures employed, as evidenced by the negligible variation in phonon dispersion. This indicates the robustness of our calculations.

The calculated values exhibit a dependency primarily on the “average behavior” across the distorted structures rather than on individual structures. This assertion is supported by the negligible impact of the distorted structures on the phonon band structure, as seen in Fig. R4. In other words, the average behavior across distorted structures has a greater impact on the calculated results than the individual variations within those structures.

Since the primitive unit cell of BaZrS₃ is itself large, we performed calculation on a 20 atoms supercell. **Following the reviewer's advice, the input file and coordinate of structure files are uploaded in git-hub (<https://github.com/pshatom/piezo-files>); this link has been provided in the Methods section. This information can be used to reproduce results.**

Fig. R4 (also included in Supplementary Information): Phonon band structure for BaZrS₃ (a) without including random displacements and (b) including random displacements. These results suggest that the structure is dynamically stable and highlights absence of imaginary phonons.

[Comment (3)] *Is the method (computing piezoelectric tensors using the average of some randomly distorted structures) used by previous work? If yes, please provide reference.*

The correlation between ionic displacements and polarization is well established in the literature (e.g., Ref. 19-21 in the main manuscript). That being said, the method employed here for computing piezoelectric tensors was devised by us specifically to address the observed piezoelectric behavior in centrosymmetric chalcogenide perovskite materials. This is an original contribution of our work.

[Comment (4)] *On the origin of piezoelectricity, can the author discuss the possibility of defect-induced piezoelectricity in BaZrS₃? How do they rule out this possibility?*

The unit cell of BaZrS₃ is comprised of 20 atoms and contains a significant volume of vacant space. This loosely packed structure of BaZrS₃ allows for an extended displacement of the ions, resulting in symmetry breaking and an enhanced dipole moment. Our first principles calculations (**Fig. 2 in the main manuscript**) confirmed that this is indeed the case.

That being said, we agree with you that defects are unavoidable in any material system and such defects could further break the symmetry and further amplify the piezoelectric response beyond what is shown in Fig. 2 for a pristine (i.e., defect-free) BaZrS₃ structure. In fact, your comment is very insightful as “defect-engineering” could be an avenue to intentionally break the centrosymmetry of chalcogenide perovskites and thus further amplify the piezoelectric response in this class of materials. **The above discussion has been included in the conclusion section of the revised manuscript as a possible “direction for future research”.** Thank you for bringing this out.

[Reviewer #3's Comments] *Abir et al. report piezoelectricity in lead-free chalcogenide perovskite BaZrS₃ using piezoresponse force microscopy measurements. Through first-principles calculations, despite BaZrS₃'s centrosymmetric structure, it's argued that such materials readily polarize under deformation due to a loosely packed unit cell with significant vacant space. This unique structure, calculations show, allows for extended ion displacement, leading to symmetry reduction and an enhanced displacement-mediated dipole moment. Under sinusoidal pressure loading, composites of BaZrS₃ particles dispersed in a polycaprolactone (PCL) matrix are shown to generate a peak areal power density of ~103.5 μW cm⁻² and are shown to harvest energy from routine body movements. The study is interesting in a highly topical area. However, there are several issues, and conclusions in their present form are not supported by experimental data.*

Thank you so much for your careful reading of our manuscript, and for your constructive criticism, advice and valuable suggestions. Your inputs have greatly improved the quality of the manuscript and we are grateful to you for your time and effort.

[Comment (1)] *What is the thickness of the BaZrS₃ films used for the PFM study? It's not mentioned in the manuscript.*

The BaZrS₃ films used for PFM study were ~300 nm in thickness. Shown below is the cross-sectional SEM image of a typical BaZrS₃ film deposited on quartz. This figure has been added in the revised supplementary information file.

Fig. R5 (also included in the Supplementary Information): Cross-sectional SEM image of a typical BaZrS₃ thin film over quartz.

[Comment (2)] PFM studies are performed on thin films of BaZrS₃ on quartz without any conductive bottom electrode. This is not a standard configuration for PFM, so, how do authors interpret their response which should be different compared to the standard configuration?

It remains a challenge to grow BaZrS₃ films over conductive substrates due to the need for high temperature (~1050 °C) synthesis in Sulfur atmosphere [1]. We attempted to grow BaZrS₃ films on a variety of metal substrates including steel and titanium, but growth was not possible. This led us to use quartz as the bottom substrate. Our PFM setup with BaZrS₃ film grown on quartz is shown in Fig. R6. In this set-up, a conductive atomic force microscopy (AFM) tip serves both as the top electrode and the sensing element. The main limitation of this set-up is that the electric field generated beneath the AFM tip is highly concentrated due to the nanoscale size of the tip, which can result in inaccuracies [2]. A number of studies in literature [3-5] have shown that this problem can be mitigated by coating the top surface of the piezoelectric film with a thin conductive [3-5] coating to reduce charge concentration. This method of top coating the piezoelectric film also generates a relatively uniform electric field and surface deformation, making it preferable for accurately detecting the piezoelectric response [6-8]. We therefore sputter coated our BaZrS₃ films with a ~5 nm layer of Pt. Given the thinness of the Pt relative to the BaZrS₃ film (~300 nm), its impact on the PFM results are considered negligible [3-8].

To prove that our set-up can discriminate between materials, in our revised submission we carried out PFM characterization for BaZrO₃ films on quartz substrate with a thin ~5 nm layer of Pt on the top surface. BaZrO₃ was chosen since based on our *ab initio* calculations, BaZrO₃ should exhibit weak piezoelectricity relative to BaZrS₃ (see Fig. 2 in the main manuscript). The test results (Fig. R7-8) indicated minimal change in piezo-response amplitude with increasing voltage for BaZrO₃, suggesting that the PFM setup was capable of detecting and differentiating between the piezoelectric responses of the materials being tested. The above discussion has been added to the **Methods section** of the revised manuscript.

Fig. R6 (also included in supplementary information): (Left) Schematic diagram of Single Frequency PFM, (Right) Configuration used in our tests.

Nonetheless, we acknowledge the limitations of this non-standard PFM setup, particularly the absence of a bottom conductive electrode, which prevents the formation of a uniform electric field through the entire thickness of the film. This configuration primarily influences the near-surface regions of the sample, potentially leading to measurements that could differ somewhat from bulk properties. Nevertheless, the primary goal of our PFM study is to demonstrate the presence of piezoelectricity in this material, which this setup effectively achieves. Moreover, the piezoelectric nature of BaZrS₃ is also corroborated by our *ab initio* calculations and the testing of bulk BaZrS₃/polymer composite devices for power harvesting.

Fig. R7 (also included in the **Supplementary Information**): BaZrO₃ thin film over quartz vertical piezoresponse force microscopy (PFM) height, phase and amplitude images at voltages of 1-6 V.

Fig. R8 (also included in Supplementary Information): PFM amplitude as a function of applied voltage showing the measured effective piezoelectric charge coefficient ($d_{33,eff}$) for BaZrO₃ vs BaZrS₃.

References cited in Response to Comment (2) of Reviewer 3:

1. Comparotto, C., Ström, P., Donzel-Gargand, O., Kubart, T. & Scragg, J. J. S. Synthesis of BaZrS₃ Perovskite Thin Films at a Moderate Temperature on Conductive Substrates. *ACS Applied Energy Materials* 5, 6335–6343 (2022).
2. Lepadatu, S., Stewart, M. & Cain, M. G. Quantification of electromechanical coupling measured with piezoresponse force microscopy. *Journal of Applied Physics* 116, 066806 (2014).
3. Güthner, P. & Dransfeld, K. Local poling of ferroelectric polymers by scanning force microscopy. *Applied Physics Letters* 61, 1137–1139 (1992).
4. Gruverman, A., Auciello, O. & Tokumoto, H. Scanning force microscopy for the study of domain structure in ferroelectric thin films. *Journal of Vacuum Science & Technology B: Microelectronics and Nanometer Structures Processing, Measurement, and Phenomena* 14, 602–605 (1996).
5. Poyato, R., Huey, B. D. & Padture, N. P. Local piezoelectric and ferroelectric responses in nanotube-patterned thin films of BaTiO₃ synthesized hydrothermally at 200 °C. *Journal of Materials Research* 21, 547–551 (2006).
6. Maiwa, H. & Ichinose, N. Measurement of Electric-Field-Induced Displacements in (Pb, La)TiO₃ Thin Films Using Scanning Probe Microscopy. *Japanese Journal of Applied Physics* 39, 5403 (2000).
7. Gerber, P. et al. Effects of the top-electrode size on the piezoelectric properties (d_{33} and S) of lead zirconate titanate thin films. *Journal of Applied Physics* 96, 2800–2804 (2004).
8. Bravina, S. L. et al. Top electrode size effect on hysteresis loops in piezoresponse force microscopy of Pb(Zr,Ti)O₃-film on silicon structures. *Journal of Applied Physics* 112, 052015 (2012).

[Comment (3)] PFM is used as the sole evidence for the piezoelectricity of the BaZrS₃ films. Obtained PFM images (Fig.1) show quite a non-uniform piezoresponse, which can arise from multiple extrinsic factors (e.g. morphological channel crosstalk, electrostatic contributions, and field-induced ion dynamics) and not just the intrinsic piezoelectricity. How do authors rule out these various extrinsic contributions and conclude that the experimental results support the suggested mechanism?

Piezoelectric measurements on the BaZrS₃ thin film were conducted using the single frequency Piezo Force Microscopy (PFM) mode (**Fig. R6**) on an Asylum Research MFP-3D system. A conductive cantilever (ASYELEC.01-R2) was used to scan over the sample surface in contact mode. The MFP-3D system features a built-in function generator and a lock-in amplifier connected to the deflection of the AFM cantilever [1]. In this setup, the conductive tip not only applies voltage to the electrode but also measures the piezoelectric motion. This conductive tip, commercially available from Oxford Instruments, is made of silicon coated with Ti/Ir, specifically designed for nano-electrical and PFM measurements. It has a nominal spring constant of 2.8 N/m, a tip radius of ~25 nm, and a free air resonance frequency of ~75 kHz. A high-stiffness cantilever was chosen to minimize the influence of electrostatic interactions on the piezoelectric measurements. The typical contact resonance frequency of this conductive tip is ~340 kHz. The frequency of the signal applied to the sample was close to this resonance peak, which is significantly higher than the low-pass cutoff frequency of the AFM topography feedback loop. A consistent tip pressure helped reduce variability in measurements caused by topographical variations.

The vertical deflection signal of the cantilever was recorded by the lock-in amplifier. By multiplying the deflection signal by the calibration constant of the photodetector sensitivity, the amplitude of the tip vibration was obtained. The process of determining calibration constant is described in response to your question 7. Imaging was performed by scanning at a 90-degree angle to ensure reproducibility. The sample surface was coated with a conducting Pt layer to reduce charge concentration and ensure a uniform electric field (**Fig. R6- right**).

A control PFM experiment was also performed using a similar setup with a BaZrO₃ thin film, which effectively showed minimal change in the amplitude of the signal with increasing voltage (**Fig. R7-R8**). This is consistent with our *ab initio* calculations (**Fig. 2** in main manuscript), which showed negligible piezo response in BaZrO₃ relative to that of BaZrS₃. These results indicate that our PFM setup is capable of detecting and differentiating between the piezoelectric behavior of the underlying material.

The above details have been included in the **Methods section** of the revised manuscript.

References cited in Response to Comment (3) of Reviewer 3:

1. Chapter 7 - Single Frequency Piezo Force Microscopy (PFM), SPM Applications Guide, User Guide 3, Revision: A-2053, Asylum Research an Oxford Instruments company 16, (2018). https://mmrc.caltech.edu/Asylum/Manuals/AR_ApplicationsGuide_16A.pdf

[Comment (4)] *Theoretical calculations show centrosymmetric BaZrS₃ is polarized under deformation due to a loosely packed unit cell with significant vacant space. In PFM measurements, however, an electric field is applied and an apparent strain response is detected. So the question arises, do theoretical calculations support electrically driven ionic displacement leading to microscopic strain that is large enough to detect?*

We appreciate your question regarding the direct and inverse piezoelectric effect. The existence of one is a thermodynamic consequence of the other. Thus, any polarization induced by an external electric field can manifest as strain, and vice versa, underscoring the connection between electric field and strain in piezoelectric materials. Yes, theoretical calculations support electrically driven displacements leading to strain as the magnitude of polarization generated in either case will remain same. By appropriately setting the electric field parameters (EFIELD) within VASP, one can perform such calculations.

[Comment (5)] *How is the magnitude of the piezoelectricity (e.g. d₃₃) expected to scale with increasing deformation?*

For small deformations, there is a near-linear relationship between the applied mechanical strain and the induced electric polarization. In this linear regime, the piezoelectric coefficients will be constant. However, for larger deformation, the response enters a ‘non-linear regime’ where the relationship between strain and polarization is no longer linear. Instead, the response will start to saturate because the alignment of dipoles due to displacement within the material reaches a saturation point, where further alignment is not possible, leading to a plateau in the response (Such plateau is evident in **Fig. 1h** and **Fig. S3** in the main paper). In our study, the piezoelectric constants were calculated for small deformations from the slope of the linear portion of the response.

[Comment (6)] *What’s the role of flexoelectricity if any in the appearance of non-zero dipole moment under deformation? This is given flexoelectricity can induce polarization even in centrosymmetric materials as has been evidenced by many reports in literature.*

Thank you for raising this important question. In response to your query, we performed detailed calculations to study the role of flexoelectricity in BaZrS₃. Flexoelectricity is a 4th order tensor described as the **polarization response to strain gradient**, where $P_i = f_{ijkl} \frac{\partial \epsilon_{kl}}{\partial x_j}$ is the polarization induced by the strain gradient, f_{ijkl} are flexoelectric tensors and $\frac{\partial \epsilon_{kl}}{\partial x_j}$ is the strain gradient. To study flexoelectricity in BaZrS₃, first-principles calculations are performed with the Vienna ab initio Simulation Package (VASP) [1]. Atomic interactions are described using the Projector Augmented Wave (PAW) method [2]. The Perdew-Burke-Ernzerhof (PBE) generalized gradient approximation (GGA) functional is employed to account for exchange-correlation effects [3]. A planewave expansion with an energy cutoff of 520 eV is utilized [4]. The Brillouin zone is sampled using a Monkhorst-Pack k-points mesh grid with a precision level of 0.03 [5].

To circumvent the issue of periodic boundary conditions in a supercell with non-uniform

strain, we construct an accordion supercell where strain gradient is sinusoidal in nature and hence preserves periodicity of the boundaries – this same approach has been used in Ref. 6–8. Consequently, the strain distribution adheres to a cosine variation, while displacements follow a sinusoidal curve: $\delta(z) = \varepsilon_{max} * \frac{h * \sin(2\pi z/h)}{2\pi}$; $\varepsilon(z) = \varepsilon_{max} * \cos(2\pi z/h)$; and $\frac{\partial \varepsilon}{\partial x} = -\varepsilon_{max} * \frac{2\pi * \sin(2\pi z/h)}{h}$. Here $\delta(z)$ is the atomic displacement in the longitudinal direction, ε_{max} is the maximum strain, z is the atom's coordinate in longitudinal direction and h is the total length of the supercell in the longitudinal direction.

Fig. R9 (also included in Supplementary Information): Displacement and strain patterns within (a) BaZrS₃ supercell of 40.16 Å length, illustrating (b) sinusoidal atomic displacements, (c) strain variation assuming a cosine profile, and (d) sinusoidal profile for strain gradient within the supercell with fractional position coordinates.

Fig. R9 shows the variation of displacement, strain, and strain gradient from our calculations on a 40.16 Å long supercell. The initiation of flexoelectricity is based on the breaking of inversion symmetry by virtue of strain gradient, while piezoelectricity is generated due to a uniform strain throughout. Hence for piezoelectricity one unit cell is sufficient to

mimic the entire system, while in case of flexoelectricity one must consider a supercell large enough to enable a converged displacement profile. Here, we demonstrate a converged atomic displacement within a 40.16 Å long supercell, consistent with previous calculations [6–8].

We apply different magnitudes of strain gradients on the relaxed supercells (Ba positions constrained) and calculate polarization at $h/4$ where the strain is zero and strain gradient is maximum, hence eliminating piezoelectric contributions. Calculations were performed for strain gradients up to $\pm 4 \times 10^7 \text{ m}^{-1}$ (**Fig. R10**) where polarization follows a linear relationship with strain, akin to the approach by Shin et al [8].

Fig. R10 (also included in Supplementary Information): Polarization as a function of strain gradient. The slope of the curve is the longitudinal flexoelectric coefficient.

The longitudinal flexoelectric coefficient, derived from the slope of the curve in **Fig. R10**, is very small ($\sim 0.0193 \text{ nC/m}$). This value, determined at the point where strain is zero and strain gradient is maximum, indicates a negligible contribution of flexoelectricity to polarization generation or the emergence of a non-zero dipole moment during deformation. **Consequently, this result suggests that the role of flexoelectricity in generation of polarization or non-zero dipole moment under deformation is minimal for BaZrS₃.**

Another **independent evidence** that the flexoelectric effect is negligible in BaZrS₃ comes from the testing of the PCL/BaZrS₃ composite films (**Fig. 4** in the main manuscript). Flexoelectricity requires a large strain gradient to polarize the material. Experimental studies that report flexoelectricity [9-10] typically involve 3-point bending in order to generate strain gradients. However, in our case, a uniform pressure load is applied along the thickness direction resulting in a uniform compression of the material (see schematic in **Fig. R11-left**). Even though there are no large strain gradients, a pronounced power generation is reported in our material. **Fig. R11-right (Fig. 4i in main paper)** compares the power harvesting ability of the

BaZrS₃-PCL film with the available literature for **PCL composites with various state-of-the-art piezoelectric additives**. Clearly the BaZrS₃ additive outperforms other well-known piezoelectric material additives, which builds confidence that this is a piezoelectric and not a flexoelectric effect.

The above results have been included in the **Revised Supplementary Information**.

Fig. R11: (Left) Schematic showing application of uniform pressure load on the polymer composite film. (Right) Areal power density comparison of BaZrS₃/PCL with the available literature for PCL films.

References cited in Response to Comment (6) of Reviewer 3:

1. J. Hafner, Ab-initio simulations of materials using VASP: Density-functional theory and beyond, *J Comput Chem* 29 (2008) 2044–2078. <https://doi.org/10.1002/JCC.21057>.
2. G. Kresse, D. Joubert, From ultrasoft pseudopotentials to the projector augmented-wave method, *Phys Rev B* 59 (1999) 1758. <https://doi.org/10.1103/PhysRevB.59.1758>.
3. J.P. Perdew, K. Burke, M. Ernzerhof, Generalized Gradient Approximation Made Simple, *Phys Rev Lett* 77 (1996) 3865. <https://doi.org/10.1103/PhysRevLett.77.3865>.
4. M. Methfessel, A.T. Paxton, High-precision sampling for Brillouin-zone integration in metals, *Phys Rev B* 40 (1989) 3616. <https://doi.org/10.1103/PhysRevB.40.3616>.
5. H.J. Monkhorst, J.D. Pack, Special points for Brillouin-zone integrations, *Phys Rev B* 13 (1976) 5188. <https://doi.org/10.1103/PhysRevB.13.5188>.
6. J. Hong, G. Catalan, J.F. Scott, E. Artacho, The flexoelectricity of barium and strontium titanates from first principles, *Journal of Physics: Condensed Matter* 22 (2010) 112201. <https://doi.org/10.1088/0953-8984/22/11/112201>.
7. A. Plymill, H. Xu, Flexoelectricity in ATiO₃ (A = Sr, Ba, Pb) perovskite oxide superlattices from density functional theory, *J Appl Phys* 123 (2018). <https://doi.org/10.1063/1.5018405>.
8. Y.-H. Shin, A. Ali, H.J. Kim, T.H. Kim, Theoretical estimation of longitudinal flexoelectric response in polar perovskite oxides, *Res Sq* (2022). <https://doi.org/10.21203/RS.3.RS-2201998/V1>.
9. C. A. Mizzi, B. Guo & L. D. Marks, Experimental determination of flexoelectric coefficients in SrTiO₃, KTaO₃, TiO₂, and YAlO₃ single crystals, *Physical Review Materials* 6, 055005 (2022).
10. P. Zubko, G. Catalan, A. Buckley, P.R.L. Welche & J.F. Scott, Strain-Gradient-Induced Polarization in SrTiO₃ Single Crystals, *Physical Review Letters* 99, 167601 (2007)

[Comment (7)] *How is the calibration done of the PFM response presented in Fig 1h? PFM images Fig1c-g have arbitrary units, however, Fig. 1h shows units of pm for piezoresponse displacement. Moreover, the measurements are performed at resonance, which should be deconvoluted to get true intrinsic response. Authors should provide more information on this.*

Regarding the calibration of PFM data, we adhered to the GetReal™ calibration procedure as outlined in the Application Guide for the Asylum research MFP-3D system [1]. This method initially conducts a thermal measurement to establish the quality factor (Q) and resonance frequency (f_0) of the cantilever. Subsequently, the "New" Sader method [2] is utilized to calculate the cantilever's spring constant (k). With this known spring constant, the thermal noise method [3] is utilized to calculate the inverse optical lever sensitivity (InvOLS). InvOLS measures how the AFM's photodetector responds to cantilever bending. Using the InvOLS feature, we can convert the voltage-induced displacement signals (in volts) into actual physical displacement (in picometers). To test the accuracy and calibration of our setup, we used Periodically Poled Lithium Niobate (PPLN), a standard PFM sample known for its consistent piezoresponse. Following the above procedure, PFM data for BaZrS₃ initially captured in arbitrary units (**Fig. 1c-g**), reflecting the raw detector signal, was subsequently converted into picometers. Details regarding calibration are now included in the **revised Methods section**.

The Asylum Research MFP-3D AFM is equipped with advanced capabilities for high-resolution PFM, featuring integrated software and hardware that enhance and deconvolute piezo-response signals. The first step in our measurement process involves accurately identifying the cantilever's resonance frequency using a preliminary frequency sweep. This identifies the frequency at which the maximum amplitude response occurs, signaling the cantilever's resonance. At this frequency, the system measures both the amplitude and phase of the cantilever deflection, which are influenced by the piezoelectric displacement of the sample surface in response to the applied voltage. An integrated lock-in amplifier isolates the piezo-response signal from noise, synchronizing with the oscillating voltage at the resonance frequency to measure the response specifically associated with the driving frequency [4]. Regular calibration of the PFM setup is essential, utilizing reference materials with well-documented piezoelectric constants to establish a baseline for converting voltage-induced displacement signals into quantifiable piezoelectric coefficients. For this purpose, we used a periodically poled lithium niobate standard PFM sample from Bruker AFM Probes.

References cited in Response to Comment (7) of Reviewer 3:

1. Chapter 20 - Spring Constant Calibration, SPM Applications Guide, User Guide 3, Revision: A-2053, *Asylum Research an Oxford Instruments company*, **16** (2018).
2. Sader, J. E. et al. Spring constant calibration of atomic force microscope cantilevers of arbitrary shape. *Review of Scientific Instruments* **83**, 103705 (2012).
3. Walters, D. A. et al. Short cantilevers for atomic force microscopy. *Review of Scientific Instruments* **67**, 3583–3590 (1996).
4. Jesse, S., Mirman, B. & Kalinin, S. V. Resonance enhancement in piezoresponse force microscopy: Mapping electromechanical activity, contact stiffness, and Q factor. *Appl. Phys. Lett.* **89**, 022906 (2006).

[Comment (8)] For practical energy harvesting and power density demonstration, BaZrS₃ powder is mixed with a polymer PCL, which is a weakly piezoelectric material. It would be ideal to conduct these experiments with a non-piezoelectric polymer in order to ensure that the generated response is more or less coming entirely from BaZrS₃ and not from the polymer. From the presented data, it is difficult to draw a definitive conclusion

Following your advice, we have repeated the experiments with Polymethyl Methacrylate (PMMA) as the polymer matrix. Open circuit voltage (V_{oc}) vs. time plots are shown for ~0.35 mm thick PMMA films in the 1.0–4.0 Hz frequency range at 30 PSI pressure (**Fig. R12**). The test data indicates that PMMA shows negligible piezoelectric response. Under identical test conditions the PMMA/BaZrS₃ composite film (~10 wt% BaZrS₃ loading), exhibits a pronounced maximum V_{oc} of ~62.4 V, which confirms that the generated response is predominantly from BaZrS₃ and not from the polymer.

Fig. R12 also shows a comparison of PMMA/BaZrS₃ (~10 wt% BaZrS₃ loading) with the weakly piezoelectric PCL and PCL/BaZrS₃ (~10 wt% BaZrS₃ loading) films, with the maximum V_{oc} recorded as ~31.6 V and ~99.6 V, for PCL and PCL/BaZrS₃, respectively. Clearly the presence of BaZrS₃ additives greatly enhance the piezoelectric performance of the baseline PCL material. These results demonstrate the effectiveness of BaZrS₃ in boosting the piezoelectric performance of polymeric systems into which they are injected.

These results have been included in the revised manuscript.

Fig. R12 (also included in the Supplementary Information): Open circuit voltage response of PMMA polymeric film for 1.0–4.0 Hz load frequency, PMMA/BaZrS₃ (~10 wt%) composite film, PCL film, and PCL/BaZrS₃ (~10 wt%) composite film for ~4.0 Hz load frequency. All samples were tested at ~30 PSI applied pressure.

[Comment (9)] *Why do the V_{oc} and I_{sc} increases up to ~15% $BaZrS_3$ weight fraction, and decrease with a further increase in the weight fraction of $BaZrS_3$? Significant agglomeration/voids seem to happen at ~30% $BaZrS_3$ weight fraction.*

The reason behind the loss of piezoelectric performance at high loadings (i.e., above ~15 wt%) is evident in the backscattered electron detector images shown in **Fig. R13**. At high loading fractions, we find that there is significant agglomeration of $BaZrS_3$ particles which leads to a weak particle-matrix interface. This leads to particle debonding (shown schematically in **Fig. R14**). Such debonded particles are electrically and mechanically disconnected from the polymer network (**Fig. R14**) and hence do not contribute to the piezo-response. This explains the degradation in performance at the higher loadings.

The above discussion has now been included in the text of the revised manuscript.

Fig. R13 (Fig. 3h-j in the main manuscript): Backscattered electron detector (BSD) images from SEM of PCL- $BaZrS_3$ composite films at 5, 15 and 30 wt.% loading respectively.

Fig. R14: Debonding of agglomerated $BaZrS_3$ particles at high loading fractions leading to loss of performance.

[Comment (10)] *What role does triboelectricity play in the generated response, if not much, how do authors rule that out?*

The experimental setup involves a composite film made out of polycaprolactone (PCL) and BaZrS₃, while copper tape was attached on both side of the film to act as electrodes. Note that there is no gap between the copper tape and PCL/BaZrS₃ composite film during pneumatic tapping motion by the plunger. Unlike triboelectric nanogenerators that rely on the contact and separation of two triboelectric layers to generate voltage [1-3], piezoelectricity is characterized by the generation of electric charge in response to mechanical strain. The absence of two different triboelectric layers and the absence of separation or gap between copper tape and composite film rule out the possibility of triboelectric effects.

This has now been clarified in the text of the revised manuscript.

References cited in Response to Comment (10) of Reviewer 3:

1. Piezoelectric Nanogenerators Based on Zinc Oxide Nanowire Arrays. *Science* (2006) 312:242–246; doi:10.1126/science.1124005
2. Triboelectric nanogenerators as a new energy technology: From fundamentals, devices, to applications, *Nano Energy* (2015) 14:126-138. doi.org/10.1016/j.nanoen.2014.11.050
3. Triboelectric nanogenerators as flexible power sources, *npj Flexible Electronics* (2017) 1:10; doi:10.1038/s41528-017-0007-8

REVIEWERS' COMMENTS

Reviewer #1 (Remarks to the Author):

The authors have addressed my concerns and provided substantial additional evidence, which has helped improve the manuscript's quality. I therefore recommend the publication of the manuscript.

Reviewer #2 (Remarks to the Author):

All the comments have been addressed well and the manuscript has been revised appropriately. Therefore, I would recommend to accept the manuscript in its current form.

Reviewer #3 (Remarks to the Author):

The authors have carefully addressed my concerns.

It is quite interesting to see that the phonon dispersion of BaZrS₃ is not significantly affected by the small distortions.

I recommend this paper.

Only a minor suggestion:

The authors mentioned that the impact of defects on the piezoelectricity of BaZrS₃ could be a "direction for future research." Therefore, I recommend the authors cite some related works, for example. Several favorable intrinsic defects are present in BaZrS₃ [Sci. China Mater. 64, 2976–2986 (2021).

<https://doi.org/10.1007/s40843-021-1683-0>; [10.1021/acs.chemmater.5b04213](https://doi.org/10.1021/acs.chemmater.5b04213)] and their relation with inversion symmetry breaking has been studied [Phys. Rev. B 102, 205308 (2022)

DOI:<https://doi.org/10.1103/PhysRevB.102.205308>].

RESPONSE LETTER

[Reviewer #1's Comments]: *The authors have addressed my concerns and provided substantial additional evidence, which has helped improve the manuscript's quality. I therefore recommend the publication of the manuscript.*

Thank you.

[Reviewer #2's Comments] *All the comments have been addressed well and the manuscript has been revised appropriately. Therefore, I would recommend to accept the manuscript in its current form.*

Thank you.

[Reviewer #3's Comments] *The authors have carefully addressed my concerns. It is quite interesting to see that the phonon dispersion of BaZrS₃ is not significantly affected by the small distortions. I recommend this paper.*

Only a minor suggestion:

The authors mentioned that the impact of defects on the piezoelectricity of BaZrS₃ could be a "direction for future research." Therefore, I recommend the authors cite some related works, for example. Several favorable intrinsic defects are present in BaZrS₃ [Sci. China Mater. 64, 2976–2986 (2021). <https://doi.org/10.1007/s40843-021-1683-0>; 10.1021/acs.chemmater.5b04213] and their relation with inversion symmetry breaking has been studied [Phys. Rev. B 102, 205308 (2022) DOI:<https://doi.org/10.1103/PhysRevB.102.205308>].

Thank you for pointing out these references. They have been cited in the revised manuscript as References 38-40.